# Marine Actinomycetes-Derived Secondary Metabolites Overcome TRAIL-Resistance via the Intrinsic Pathway through Downregulation of Survivin and XIAP

**DOI:** 10.3390/cells9081760

**Published:** 2020-07-22

**Authors:** Mohammed I. Y. Elmallah, Sheron Cogo, Andrei A. Constantinescu, Selene Elifio-Esposito, Mohammed S. Abdelfattah, Olivier Micheau

**Affiliations:** 1LNC, INSERM, UMR1231, F-21079 Dijon, France; sheron_cogo@hotmail.com (S.C.); andrei.ac@windowslive.com (A.A.C.); 2UFR Science de Santé, Université de Bourgogne Franche-Comté, F-21079 Dijon, France; 3Chemistry Department, Faculty of Science, Helwan University, 11795 Ain Helwan, Cairo 11795, Egypt; mabdelfattah@science.helwan.edu.eg; 4Graduate Programme in Health Sciences, Pontifícia Universidade Catolica do Parana, Curitiba 80215–901, Parana, Brazil; seleneesposito@gmail.com; 5Marine Natural Products Unit (MNPRU), Faculty of Science, Helwan University, 11795 Ain Helwan, Cairo 11795, Egypt

**Keywords:** TRAIL, marine actinomycetes, apoptosis, therapy

## Abstract

Resistance of cancer cells to tumor necrosis factor-related apoptosis-inducing ligand (TRAIL)-induced apoptosis represents the major hurdle to the clinical use of TRAIL or its derivatives. The discovery and development of lead compounds able to sensitize tumor cells to TRAIL-induced cell death is thus likely to overcome this limitation. We recently reported that marine actinomycetes’ crude extracts could restore TRAIL sensitivity of the MDA-MB-231 resistant triple negative breast cancer cell line. We demonstrate in this study, that purified secondary metabolites originating from distinct marine actinomycetes (sharkquinone (1), resistomycin (2), undecylprodigiosin (3), butylcyclopentylprodigiosin (4), elloxizanone A (5) and B (6), carboxyexfoliazone (7), and exfoliazone (8)), alone, and in a concentration-dependent manner, induce killing in both MDA-MB-231 and HCT116 cell lines. Combined with TRAIL, these compounds displayed additive to synergistic apoptotic activity in the Jurkat, HCT116 and MDA-MB-231 cell lines. Mechanistically, these secondary metabolites induced and enhanced procaspase-10, -8, -9 and -3 activation leading to an increase in PARP and lamin A/C cleavage. Apoptosis induced by these compounds was blocked by the pan-caspase inhibitor QvD, but not by a deficiency in caspase-8, FADD or TRAIL agonist receptors. Activation of the intrinsic pathway, on the other hand, is likely to explain both their ability to trigger cell death and to restore sensitivity to TRAIL, as it was evidenced that these compounds could induce the downregulation of XIAP and survivin. Our data further highlight that compounds derived from marine sources may lead to novel anti-cancer drug discovery.

## 1. Introduction

The marine environment encompasses a great microbial biodiversity. These microorganisms are able to survive in harsh environmental conditions including high salinity, temperature, and pressure [1] and are currently considered as the major producer of a great variety of pharmacologically active secondary metabolites. Among them are actinomycetes, particularly *Streptomyces* sp, that represent the storage pool for bioactive compounds with antibacterial, antiviral, anti-inflammatory, antimalarial, or antitumor activity [2,3,4,5].

The critical hallmark features of cancer initiation and progression are primarily associated with the ability of living cells to escape apoptosis and undergo uncontrolled proliferation [6]. Currently, the standard treatment of solid tumor includes surgery to remove the cancerous lump followed by chemo/radiotherapies to get rid of the residual cells. The main obstacle of these conventional therapies is their unspecific mode of action, as they also target normal cells. Efforts are now directed towards developing lead drugs that can kill tumor cells, but spare healthy cells. This can be achieved through restoration of tumor cell sensitivity to various apoptotic signaling pathways or by targeting pro-apoptotic receptors of the tumor necrosis (TNF) superfamily such as TRAIL-R1 or TRAIL-R2 [7,8]. 

Tumor necrosis factor-related apoptosis-inducing ligand (TRAIL) is a type II transmembrane ligand belonging to the TNF family, like TNF-α or Fas/APO-1 (CD95). TRAIL was found to induce p53-independent apoptotic cell death in different cancer cell lines without affecting healthy cells, which makes TRAIL a promising therapeutic agent in oncology [9,10,11]. This ligand can bind to five distinct cell-surface receptors. Two of them harbor the conserved intracellular death domain (DD), TRAIL-R1/DR4, and TRAIL-R2/DR5, allowing them to transduce apoptosis upon TRAIL binding. The remaining three include TRAIL-R3/DcR1, TRAIL-R4/DcR2, and the soluble receptor known as osteoprotegerin (OPG). These receptors, often coined decoy receptors (DcRs), either lack or harbor an incomplete DD or are expressed as soluble receptors, respectively, and are thus unable to induce apoptosis (see [12,13]). Apoptosis induced by TRAIL is thus mostly conveyed by DR4 and DR5 through the formation of a multimolecular scaffold complex called DISC, which allow recruitment of the adaptor protein, Fas-associated death domain (FADD) through homotypic interactions involving its DD and the DD of DR4 or DR5 [14,15]. FADD, in turn, thanks to its death effector domain (DED), recruits the pro-initiator caspases-8 and -10, allowing their activation by autocatalytic cleavage and subsequent release of their active fragments into the cytosol [16,17]. Provided that the initiator caspases are robustly activated, caspase-8 or -10 induce the cleavage of executioner caspases including caspase-3, allowing the dismantling of the cell by apoptosis. If the amount of activated initiator caspase is not sufficient to allow direct execution of apoptosis, through this so-called extrinsic pathway, cell death can still occur through amplification of the signal via mitochondria. This intrinsic pathway is generally activated by cellular stresses or insults, including DNA damage [15,18,19,20]. It can be activated coincidentally by TRAIL receptors in a process requiring the cleavage of the BH3-interacting domain death agonist (Bid) by caspase-8. Truncated Bid (tBid) [21], translocates to the outer mitochondrial membrane, inducing activation of the pro-apoptotic molecules Bcl-2-associated X protein (Bax) and Bcl-2 homologous antagonist killer (Bak), leading to change in the outer membrane potential and to the release of cytochrome c (cyt c) into the cytosol. Once in the cytosol, cyt c interacts with the apoptotic protease activating factor 1 (Apaf-1) and ATP, allowing the recruitment and activation of another initiator caspase, the caspase-9 within a soluble multiproteic scaffold complex; known as the apoptosome. Like caspase-8, the activated caspase-9 is able to trigger the activation of the effector caspase-3 by cleavage and thus enable proper execution of the apoptotic signal [22]. 

Despite encouraging results demonstrating that TRAIL or its derivatives exhibit clinical pro-apoptotic activity, their efficacy, so far, have lagged behind expectations [23,24]. Several factors may explain why these trials have failed to demonstrate the efficacy of TRAIL. Tumor resistance to TRAIL-induced apoptosis is probably one of the most likely explanations. Impaired pro-apoptotic signaling may result from a lack or the downregulation of the agonist receptors DR4 and DR5, from the cell-surface expression of TRAIL decoy receptors (DcR1 and DcR2) [25,26,27], from the overexpression of DISC inhibitors, such as the caspase-8 homologue cellular FLICE-like inhibitory protein (c-FLIP) [28,29], from the downregulation of the pro-initiator caspase-8, or from the overexpression of the antiapoptotic proteins B-cell lymphoma 2 (Bcl-2), B-cell lymphoma-extra-large (Bcl-xL), or inhibitors of apoptosis (IAPS) [22,24].

Expression levels or subcellular localization of these proteins are scarcely, if not, evaluated in the tumor biopsies of patients enrolled in the clinical trials aiming at evaluating the efficacy of TRAIL or its derivatives. Yet, given that TRAIL agonist receptors are generally well expressed in cancer cells, including patient biopsies, there is still hope that novel drugs, displaying the ability to restore tumor cell sensitivity to TRAIL-induced apoptosis may change the paradigm and allow the use of TRAIL in the clinic to treat patients.

Because, we and others have demonstrated that crude extracts from marine actinomycetes may represent a novel source of TRAIL sensitizers [22,30,31], the present study was conducted to investigate the TRAIL-resistance overcoming or sensitizing ability of eight marine actinomycetes-derived secondary metabolites (sharkquinone, resistomycin, undecylprodigiosin, butylcycloheptylprodigiosin, elloxazinone A and B, carboxyexfoliazone, and exfoliazone).

We provide evidence here that all compounds when combined to TRAIL display pro-apoptotic enhancing activity owing, at least in part, to their ability to induce the downregulation of the anti-apoptotic proteins survivin and XIAP. Our results represent an initial seed towards the development of lead structures in the field of cancer drug discovery from marine sources.

## 2. Material and Methods

### 2.1. Chemicals and Reagents

All chemicals and reagents used were of analytical grade and were purchased from Sigma Aldrich (St. Louis, MO, USA). Annexin V-FITC apoptosis detection kit and 7-aminoactinomycin D (7-AAD) were received from Biolegend (San Diego, CA, USA). Antibodies for immunoblotting including PARP (Santa Cruz, sc-25780), cPARP (Cell Signaling, D64E10), laminA/C (Abcam, ab133269), caspase-8 (MBL, M032-3), caspase-10 (MBL, M059-3), caspase-9 (MBL, M054-3), caspase-3 (Cell signaling, 9665), anti-DR4 (Abcam, ab8415), anti-DR5 (Abcam, ab8416), GAPDH (Santa Cruz, sc-47724), HSC70 (Santa Cruz, sc-7298), BiP-1 (Cell signaling, C50B12), IRE1 (Cell signaling, 14C10), RIPK1 (BD Bioscience, 610459), XIAP (Santa Cruz, sc-55550), and survivin (Abcam, EP2880Y). Bradford reagent was obtained from Bio-Rad (Marnes-la-Coquette, France). Chemiluminescent Western blots detection kit was purchased from Advansta (San Jose, CA, USA). 

### 2.2. Microbial Strains

Marine bacterial strains considered in the current study had previously been assigned to actinomycetes based on partial 16S rRNA and morphological appearance [31,32,33,34]. The actinobacterial strains EGY1 and EGY34 were isolated from sediment samples obtained from the Red Sea, Egypt. The strain RA2 was isolated from a sponge sample collected from Ras Mohammed, South of Sinai, Egypt. The sponge was identified as *Spheciospongia mastoidea* by Prof. Rob. W. M. van Soest (University of Amsterdam, Netherlands). The strain EG25 was isolated from a sea-sand sample obtained from Marsa Matruh city, Mediterranean Sea, Egypt.

### 2.3. Cultivation of Actinomycetes Strains

The strains EGY1, EGY34, and RA2 were grown on agar plates and inoculated into Erlenmeyer flasks containing Waksman liquid media with 50% sterilized sea water. The media contained glucose (2.0 g/100 mL), meat extract (0.5 g/100 mL), peptone (0.5 g/100 mL), dried yeast (0.3 g/100 mL), NaCl (0.5 g/100 mL) and CaCO_3_ (0.3 g/100 mL). The strain EG25 was grown on malt extract-yeast extract media containing malt extract (10.0 g/100 mL), glucose (4.0 g/100 mL), yeast extract (4.0 g/100 mL) and sea water (50%). All culture flasks of EGY1, EGY34, RA2, and EG25 were incubated in a shaker at 28 °C for 3–4 days.

### 2.4. Extraction and Isolation of Compounds ***1**–**8***

The culture broth (4.0 L) of each strain was centrifuged and extracted three times with ethyl acetate. The mycelial cake was extracted three times with acetone and combined with the extract of culture broth. The crude extract (2.4 g) of *Streptomyces* sp. EGY1 was fractionated using silica gel flash column chromatography (30 × 500 mm) with a gradient of 0–100% CH_3_OH /CH_2_Cl_2_ resulting in four fractions. Fraction II (250.6 mg) was purified by Sephadex LH-20 column chromatography (φ 25 × 240 mm) eluted with CH_3_OH followed preparative HPLC (YMC-Pack ODS-AM, 10 × 250 mm; eluent, 65% MeOH; flow rate, 2.0 mL/min; UV detection at 254 nm; YMC Co Ltd, Kyoto, Japan) to give 5.30 mg of sharkquinone (**1**). The brown crude extract (2.60 g) of *Streptomyces* sp. EGY34 was dissolved in methanol and left to stand overnight. A yellow precipitate separated out of the solution was observed. The precipitate was filtrated and washed three times with methanol to give 30.6 mg of resistomycin (**2**) as a yellow solid. The crude extract of the actinomycete RA2 (3.52 g) was fractionated using silica gel column chromatography through a stepwise gradient solvent system of increasing polarity (CH_2_Cl_2_: CH_3_OH = 100:0, 95:5, 90:10, 85:15, 80:20, 70:30, 60:40, 0:100) to obtain four fractions. The red bands in fraction I (62.31 mg) was purified by preparative thin layer chromatography (4 plates, 20 × 20 cm, CH_2_Cl_2_/5% CH_3_OH) to give 4.20 mg of undecylprodigiosin (**3**) and 5.30 mg of butylcycloheptylprodigiosin (**4**) as red solids. For EG25, the crude extract (3.70 g) was applied to a silica gel column (45 × 2.6 cm) to give five fractions. Fraction II (60.41 mg) was purified by PTLC (5 plates, 20 × 20 cm, CH_2_Cl_2_/CH_3_OH 95:5) to obtain 1.9 mg of elloxazinone A (**5**). After chromatographic separation of fraction III (460.4 mg), the exfoliazone (**7**, 2.91 mg) precipitated from the chloroform/methanol solution on slow evaporation. Considering fraction IV, 7.80 mg of carboxyexfoliazone (**8**) was isolated by PTLC (3 plates, 20 × 20 cm, CH_2_Cl_2_/CH_3_OH 90:10) followed by Sephadex LH-20 (MeOH). The fraction V (943.8 mg) containing the polar elloxazinone B (**6**) was re-chromatographed on Sephadex LH 20 (CH_2_Cl_2_/40% CH_3_OH) to give three sub-fractions Va (456.1 mg), Vb (312.5 mg) and Vc (170.2 mg). Elloxazinone B (**5**, 8.31 mg) was obtained from sub-fractions Vc.

### 2.5. Cell Culture

Human colon cancer HCT116 (CCL-247) and Jurkat (TIB-152) cell lines were obtained from American Type Culture Collection (ATTC). The human MDA-MB-231 breast cancer cell line was a kind gift of Dr Patrick Legembre (Rennes, France). Jurkat-deficient for FADD or Caspase-8 was provided by Dr Juo and Blenis [35,36]. HCT116 and MDA-MB-231 cells were cultured in Dulbecco’s modified Eagle’s medium (DMEM). Jurkat cells were grown in RPMI1640 medium. Both media were supplemented with 4.5 g/L of glucose, 4 mmol/L of L-glutamine, and 10% heat-inactivated fetal calf serum (FCS). Cells were maintained in 5% CO_2_ at 37 °C.

### 2.6. CRISPR-Mediated Gene Deletion

Caspase-8-deficient HCT116 cells were generated using the pLC-RFP657-CASP8 plasmid, a gift from Dr Beat Bornhauser (Addgene plasmid #75164; http://n2t.net/addgene:75164; RRID: Addgene_75164) [37]. Single clones were FACS-sorted and screened for the loss of the caspase-8 by Western blot and loss of sensitivity to TRAIL- and FasL-induced apoptosis.

### 2.7. Cell Viability Assay

The in vitro cytotoxicity of the marine secondary metabolites (**1**–**8**) was measured by methylene blue assay as previously described [30]. In brief, HCT116 and MDA-MB-231 cells were cultured at 4 × 10^4^ cells/well in flat-bottomed 96 well plates and then treated with various concentrations of the secondary metabolites (sharkquinone, resistomycin, undecylprodigiosin, butylcycloheptylprodigiosin, elloxazinone A and B, carboxyexfoliazone, and exfoliazone) and incubated at 37 °C for 24 h. Following incubation, the culture media was removed, and the cells were washed three times with 100 µL PBS. Cells remaining attached to the bottom of the plate were fixed with 100 μL of 70% ethanol for 15 min. Ethanol was drained off and the plates left to dry at room temperature for 1 h and then 100 μL of methylene blue dye was added. The plates were incubated at room temperature for 15 min. The excess dye was removed by washing the plates three times under tap water followed by incubation at 37 °C for 2 h. For measurement, the dye was eluted from the fixed cells with 100 μL of 0.1 M HCl and left for 5 min at room temperature and coloration was quantified using an ELISA reader at 630 nm. The percentage of cell viability was calculated as follow:(1)$$The\;\%\;of\;cell\;viability=\frac{OD\;of\;treated\;cells}{OD\;of\;control\;cells}\times 100\%$$

### 2.8. Hoechst 33342 Staining

Nuclear morphological changes of apoptotic cells were evaluated by DNA staining with Hoechst 33342. As previously described [38], 2 × 10^5^ cells were seeded in a 12-well plate and pretreated with the compounds at final concentration of 50 µM for 6 h at 37 °C. The cells were then washed with PBS for 5 min at room temperature and fixed with 4% paraformaldehyde (PFA) for 30 min at room temperature. Afterwards, the cells were washed with PBS to remove the fixing solution and then stained with Hoechst for 15 min at 37 °C. Finally, the cells were mounted on cover slips and imaged using fluorescent microscope (Olympus, BX60) equipped with a Nikon camera (Tokyo, Japan).

### 2.9. Flow Cytometry

In a 6-well plate, 5 × 10^5^ cells/well were seeded and treated with the investigated compounds at final concentration of 50 µM with and without 250 ng/mL TRAIL and then incubated for 24 h at 37 °C. Following incubation, cells were washed with PBS and non-specific binding sites were saturated with 3% bovine serum albumin (BSA) in PBS for 30 min on ice bath. Cells were next washed with PBS followed by double staining with Annexin V and 7-AAD apoptosis detection kit according to the manufacturer’s instructions. Flow cytometric analysis was conducted using a FACScanto II flow cytometer (BD Becton-Dickinson, Franklin Lake, NJ, USA).

### 2.10. Western Blot Analysis

Cells were seeded in 6-well plates at 1 × 10^7^ cells/well and treated with the investigated compounds **1**–**8** (50 µM) in the presence and in the absence of TRAIL (250 ng) for 24 h at 37 °C. Afterwards, the cells were collected and lysed in lysis buffer (1% NP-40, Tris-HCl, 3 M NaCl, 5% glycerol). The protein expression profile was assessed by loading equal amounts of proteins (20–30 µg) from the prepared cell lysate on 12% sodium dodecyl sulfate-polyacrylamide gel electrophoresis (SDS-PAGE). The gel was then electroblotted on nitrocellulose (NC) membrane (Thermo Fisher Scientific, GE Healthcare Amersham, UK). The NC membranes were blocked with 5% non-fat milk dissolved in PBS (1×) and 0.05% Tween 20 (PBST) for 1 h at room temperature. To remove the excess milk, the membranes were washed 3 times with PBST and incubated with the protein-targeted primary antibody for overnight at 4 °C. The membranes were washed 3 times with PBST and then incubated with the corresponding horse radish peroxidase (HRP)-conjugated secondary antibody for 1 h at room temperature. Finally, the membranes were washed as previously mentioned and the signal was developed by chemiluminescent Western blots detection kit and visualized using the Bio-Rad imaging system. 

### 2.11. Statistical Analysis

Statistical analysis was performed using the non-parametric analysis of variance (two ANOVA) with Bonferonni post hoc multiple comparison test. *p* value * *p* < 0.05, ** *p* < 0.01, *** *p* < 0.001, and **** *p* < 0.0001 were considered significant. All statistical analyses were performed using Prism 5.0a software (GraphPad Software, San Diego, CA, USA). 

## 3. Results

The microorganisms used in this study include the actinobacteria strains EGY1, EGY34, RA2 and EG25 (Figure 1A). These strains were isolated from the Egyptian marine environments of both Red and Mediterranean seas. The marine isolates EGY1 was obtained from Sharks Bay, Red Sea, Egypt. It was characterized as *Streptomyces* sp. based on partial 16S rRNA gene sequences [39] and morphological characters [40]. Analysis of the crude extract by TLC showed the presence of a red band that gave a blue color reaction after spraying with 2N NaOH indicating the presence of anthraquinone moiety. Working up of the strain led to the isolation of an ana-quinonoid tetracene derivative named sharkquinone (**1**). The structure of compound **1** (Figure 1B) was determined using spectral analysis including 1D NMR [31]. The strain EGY34 was isolated from a sediment sample collected from Ras Mohammed, South of Sinai, Egypt. It was identified as *Streptomyces* sp. based on partial 16S rRNA gene sequences. The crude extract of *Streptomyces* sp. EGY34 media delivered the yellow resistomycin (**2**), that gave a fluorescence band at 365 nm and a reddish-brown coloration after staining with anisaldehyde/sulfuric acid [41]. The structure of resistomycin (Figure 1B) was determined using mass and NMR spectroscopic analysis and confirmed by comparison of these spectral data with those in the literature [42]. The actinomycete RA2 was isolated from the marine sponge *Spheciospongia mastoidea*, which was collected from Ras Mohammed protectorate, South of Sinai, Egypt. Identification of RA2 was carried out by morphological and microscopic examination [40]. The crude extract of RA2 showed on TLC (CH_2_Cl_2_/5% CH_3_OH) two reddish-pink bands. These bands gave a yellow color after treatment with 2N NaOH and blue color after spraying with anisaldehyde/H_2_SO_4_. Following extraction and evaporation of solvent, separation of the crude extract resulted in compounds (**3**) and (**4**). The chemical structures of both compounds (Figure 1B) were found to be undecylprodigiosin (**3**) and butylcycloheptylprodigiosin (**4**) based on NMR (1H and ^13^C NMR) and mass spectra [34]. Additionally, the structures were confirmed by comparison of their NMR data with the literature [43,44]. The strain EG25 was identified as actinomycete according to its morphological criteria. Analysis of the crude extract of EG25 by TLC showed the presence of several yellow bands that gave a dark brown color after spraying with the Dragendorff’s reagent. Following different chromatographic separation steps [32], the crude extract of EG25 gave compounds (**5**–**8**). The isolated compounds were identified as elloxazinones A and B (**5** and **6**), exfoliazone (**7**) and carboxyexfoliazone ((**8**) see (Figure 1B)) by spectroscopic analysis and by comparison of their spectroscopic data with those published in literatures [45,46,47].

The in vitro cytotoxicity of the marine-derived compounds (**1**–**8**) against both colorectal (HCT116) and breast (MDA-MB-231) cancer cell lines was investigated. Cancer cells were treated with variable concentrations of the identified compounds and incubated for 24 h at 37 °C. As demonstrated using the methylene blue assay, these compounds inhibited in a concentration-dependent manner the cell viability of both cell lines (Figure 2). Estimated IC_50_ for these compounds, in these experimental conditions, ranged from 7.6 to 285.8 micro molar (Table 1). Resistomycin (**2**) and undecylprodigiosin (**3**) showed the highest cytotoxic effect in both cell lines with IC_50_ of approximately 7.6 and 21.3 µM for HCT116 and 19.8 and 17.5 µM for MDA-MB-231, respectively. It should be noted here that the IC_50_ calculated from the methylene blue experiments likely overestimated the cell killing activity of these compounds since both annexin (Figure 3) and Hoechst staining (Figure 4 and Figure 5) indicate that at 50 µM they induce 15 to 50% apoptosis in MDA-MB-231 and HCT116 cells, respectively. We therefore selected this concentration for further experiments. Analysis of pro- and anti-apoptotic protein expression levels by immunoblot, 24 hours after stimulation at 50 µM, revealed that like TRAIL, all compounds were able to induce the appearance of p41 and p50 cleaved Lamin A/C and p89 kDa PARP products in cell extracts of stimulated HCTT16 cells (Figure 6A). These products are usually associated with the execution phase of apoptosis. Consistent with these findings, several initiator caspases, belonging both to the extrinsic or intrinsic pathways, as well as executioner caspases appeared to be activated by these compounds. Accordingly, much like TRAIL itself, these compounds induced the activation, as shown by their disappearance, of initiator and executioner caspases, such as caspase-8 and caspase-3, respectively. With compounds 1 to 4 inducing stronger caspase-3, caspase-8, and caspase-9 disappearance than other compounds (Figure 6A). Of note, albeit for an unknown reason, some compounds such as **1**, **3**, **4**, **6**, and **8** induced an increase of caspase-10 expression levels. Further supporting the conclusion that these compounds are likely to induce a pro-apoptotic cell death in HCT116 cells was the finding that most of them induce a reduction of both survivin and to a lesser extent of X-linked inhibitor of apoptosis protein (XIAP) protein content in these samples (Figure 6A). In line with their potential ability to trigger apoptosis, the pan-caspase inhibitor (QvD) prevented both the appearance of Lamin A/C cleaved products (Figure 6B) and apoptosis (Appendix A), induced both by these compounds and TRAIL. It should be stressed here that at least three molecules, including compounds **1**, **2**, and **3** display autofluorescence at above 450 nm, regardless of the cell line (Appendix A). Keeping in mind that apoptosis is usually detected by flow cytometry using a dual staining based on annexin V-FITC and 7AAD, and given that the detection channels for these labels correspond to 530 and 595 nm (Figure 3A), monitoring apoptosis with these compounds requires particular care. Notwithstanding, this approach can be used as demonstrated in HCT116 cells stimulated with compounds alone or combined with TRAIL (Appendix A). We thus used this assay to determine whether FADD and caspase-8 may be involved in the signal transduction machinery induced by some of these compounds to trigger apoptosis. To address this issue, we used the Jurkat cells proficient or deficient for caspase-8 or FADD, generated by Dr Juo and Blenis [35,36]. In this T lymphoma cell line, the loss of FADD or caspase-8, two essential components of the extrinsic apoptotic pathway, hardly impaired apoptosis induced by marine actinomycetes-derived secondary metabolites, with the exception of elloxazinone A, compound 5 (Figure 3A). To determine whether this finding could be generalized to other cell types, we generated another caspase-8 deficient cell line using the CRISPR/CAS9 approach (Figure 3B) and took advantage of the TRAIL-deficient cells generated previously in HCT116 cells [48], to assess whether a loss of caspase-8 would also prevent apoptosis induced by compound 5 in HCT116 cells. Whereas a deficiency in caspase-8 in HCT116 cells protect the latter from apoptosis induced by TRAIL as expected (Figure 3B), apoptosis induced by elloxazinone A (**5**) was not compromised in these cells (Figure 3C). However, Caspase-8-deficiency in these cells impaired, to some extent, apoptosis induced by elloxazinone B (**6**), carboxyexfoliozone (**7**), and exfoliazone (**8**). Loss of TRAIL-R1/TRAIL-R2 in HCT116 cells, on the other hand, except for sharkquinone (1), did not impair secondary metabolite’s pro-apoptotic potential (Figure 3C). These findings are inconsistent with the possibility that the extrinsic pathway acts as a primary trigger for apoptosis induced by these secondary metabolites.

In order to avoid bias due to the fluorescent properties of some of our compounds and with the aim of evaluating whether the latter may nonetheless enhance TRAIL-induced apoptosis, like the actinomycetes EGY1 and EGY34 crude extracts [30] from which some of the secondary metabolites were extracted, we quantified apoptosis using Hoechst dye. As shown in Figure 4, all compounds, at 50 µM were able to alone trigger the formation of apoptotic bodies, 6 hours after stimulation as visualized by chromatin condensation under the microscope (Figure 4A,B).

To assess the synergy, Jurkat and HCT116 cells were treated simultaneously with the compounds at final concentration of 50 µM and TRAIL (250 ng/mL) for 6 h. As shown in Figure 4, all compounds (**1**–**8**) were able to enhance TRAIL-induced cell death. Higher apoptotic rates were observed as a result of the combined treatment compared to single treatments (Figure 4). In Jurkat cells, and in the above-mentioned settings, with the exception of sharkquinone (**1**), whose combined effect was less efficient than the addition of the single treatments, the other compounds added to synergistic effects (Figure 4). For instance, with the exception of sharkquinone (**1**), resistomycin (**2**), and undecylprodigiosin (**3**), apoptosis induced by TRAIL combined with compounds 5 to 8 was always more superior than mere addition of single agents in HCT116 cells (Figure 4).

Consistent with these findings, immunoblot analysis indicated that the combined treatment induced strong cleavage of PARP or lamin A/C, associated with activation of initiator caspases, including capases-8, -10 and -9, as well as caspase-3, as evidenced by the disappearance of their proform (Figure 5A,B). Moreover, a large number of compounds, in combination or not with TRAIL, induced marked downregulation in the expression profile of the anti-apoptotic proteins survivin and to a lesser extent of XIAP (Figure 5A,B).

Remarkably, in the TRAIL resistant triple negative breast cancer cell line MDA-MB-231, the TRAIL-induced apoptosis enhancing activity of all these marine-derived secondary metabolites was particularly obvious, given that in these cells, for a 6 hour stimulation time, apoptosis induced by single agents alone was low, whereas combined treatments induced strong apoptosis ranging from 30 to almost 80% (Figure 5C). In line with these findings, cleavage of PARP and lamin A/C was solely evidenced in cell extracts originating from cells stimulated with both TRAIL and secondary metabolite compounds (Figure 5D). Consistently, combined treatments induced stronger caspase activation as demonstrated by the disappearance of their proform (Figure 5D). Of interest, a number of compounds significantly impaired the expression levels of the anti-apoptotic agents survivin and, albeit to a lesser extent, XIAP, regardless of the cell line (Figure 5; Figure 6), suggesting that the ability of these compounds to trigger apoptosis and to increase that induced by TRAIL likely involves regulation of survivin and XIAP.

## 4. Discussion

While tumor necrosis factor-related apoptosis inducing ligand (TRAIL) selectively induces apoptosis of tumor cells [9,11,49,50], approximately 20% of them develop resistance to TRAIL, limiting its potential use in clinic [22,49,51]. Surprisingly, cancer cells’ refractory to conventional chemotherapy, sometimes displays high sensitivity to TRAIL-induced cell death [52,53,54]. Fortunately, although the molecular mechanisms leading to defect in TRAIL signaling are diverse [55,56], a large number of drugs, from distinct classes, are able to overcome TRAIL-resistance [55,57], and probably more, are still to be discovered. 

Marine environment covers more than 70% of the earth’s surface and encompasses a great biodiversity of unexplored microorganisms that are likely to provide an interesting source of novel bioactive secondary metabolites [58,59], including compounds displaying ability to increase apoptosis induced by TRAIL [60,61,62]. Egypt is surrounded on the east and north by the Red Sea and the Mediterranean Sea with unique microbial communities, particularly actinomycetes [30,63,64,65,66], which are distributed in a variety of marine habitats including sediments and sponges [30,63]. According to the molecular data of 16S rRNA gene sequence analysis, *Streptomyces* and *Micromonospra* sp were considered as the main producers of novel secondary metabolites that possess a significant pharmaceutical potential against chemotherapeutic-resistant cancers [58,65]. We recently provided evidence, using crude extracts, that actinomycetes’ secondary metabolite may in addition allow restoration of TRAIL sensitivity in resistant tumor cells [30].

In this study, secondary metabolites were purified from four distinct actinomycetes, isolated from sediments of the Red and Mediterranean Seas as well as from the marine sponge *Spheciospongia mastoidea* (Compounds 1–8 structures are illustrated in Figure 1). These strains produce different types of natural products such as polyketides (EGY1 and EGY34) and alkaloids (RA2 and EG25). In our previous work we reported the TRAIL-resistance overcoming activity of the crude extracts EGY1 and EGY34 in the TRAIL-resistant MDA-MB-231 cell line [30]. Sharkquinone (1) and resistomycin (2) are unusual polycyclic aromatic polyketides with ana-quinonoid tetracene moiety and a benzo[c,d]pyrene ring, respectively. Aromatic polyketides are a large group of natural products with a wide range of biological activities and pharmacological properties [67,68]. The pigmented alkaloids undecylprodigiosin (3) and butylcycloheptylprodigiosin (4), obtained from the actinomycete RA2, are characterized by a pyrrolyl pyrromethene core. Both of them can be isolated from diverse types of microbes including actinomycetes [69]. These red pigments have a wide range of biological activities including anticancer properties [70]. Compounds 5–8 (elloxazinones A and B, exfoliazone, and carboxyexfoliazone) isolated from the EG25 strain contain the phenoxazinone scaffold, which have different pharmaceutical activities such as antimicrobial, antiviral, and anticancer [32,71,72]. As per literature, phenoxazinones are a group of secondary metabolites containing tricyclic rings heterocyclised by oxygen and nitrogen atoms [73]. 

The in vitro cytotoxicity of the investigated compounds (**1**–**8** structures are illustrated in Figure 1) displayed, alone, a concentration-dependent growth suppressive effect in colon HCT116 and breast MDA-MB-231 cancer cell lines (Figure 2). Among the tested compounds, resistomycin and undecylprodigiosin exhibited the highest cytotoxicity in both cell lines. Resistomycin was also found in other studies to induce significant cytotoxicity in the hepatic carcinoma (HepG2) and cervical carcinoma (Hela) cell lines, in vitro [41], and to enhance the apoptotic effect of colcemid (CL) in human megakaryoblastic leukemia CMK-7 cells [74]. Other secondary metabolites such as sharkquinone and elloxazinones A and B were found to induce apoptosis in human gastric adenocarcinoma (AGS) cells [31] and, albeit to a lesser extent, in the hepatocellular (HepG2) and breast (MCF-7) cancer cell lines [45]. Consistent with these studies, we demonstrate here that the secondary metabolites investigated in our study were all able to trigger apoptosis, alone, as evidenced by nuclear fragmentation and annexin V positivity in HCT116, Jurkat, and MDA-MB-231 cells. How these compounds induce tumor cell killing still remains unknown. All compounds, albeit to differential extent and depending on the cell line, induced activation of initiator and executioner caspases, as evidenced by the decrease of their proforms or the appearance of caspase substrate cleaved products, including PARP or Lamin A/C. Analysis of apoptosis induced by these compounds in HCT116 or Jurkat isogenic cell lines deficient or proficient for caspase-8, FADD, or TRAIL agonist receptors, however, clearly indicated that the extrinsic pathway is unlikely to be involved in the process, since neither a deficiency in caspase-8, FADD, or TRAIL receptors was able to significantly impair their ability to trigger cell death. These secondary metabolites, more likely induce apoptosis through the intrinsic pathway. Although the molecular events involved in this process remain to be determined, we provide evidence that a large number of the secondary metabolites studied here were able to induce a drastic reduction in survivin steady-state level expression and to a lesser extent, that of XIAP. These changes were further evidenced when secondary metabolites were combined with TRAIL, likely explaining, at least in part, their ability to restore or enhance TRAIL-induced apoptosis. This was particularly obvious in the TRAIL resistant triple-negative breast cancer cell line, MDA-MB-231 [27,75,76]. Inhibition of surviving by the small molecule inhibitor (3-(2methoxyethyl)-2-methyl-4,9-dioxo-1-(pyrazin-2-ylmethyl)-4,9-dihydro-3H-naphtho(2¨C-d)-imidazol-1-ium bromide) also known as YM155, was found to sensitize MDA-MB-231 cells to TRAIL [77]. Of interest, YM155 was also found by an independent team to sensitize other triple-negative breast cancer cell lines to CD34+ cells engineered to express TRAIL on their cell surfaces [78]. In our previous study, we also found that sharkquinone was able to restore the TRAIL-induced apoptosis in the stomach gastrocarcinoma cell line AGS [33]. Other studies reported the anti-cancer effect of resistomycin as a single agent against different cancer cell lines [41,74] and its ability to restore TRAIL-induced apoptosis in resistant AGS cells [79]. We report, here, for the first time, the TRAIL-resistance overcoming activity of undecylprodigiosin, butylcycloheptylprodigiosin, elloxazinones A and B, carboxyexfoliazone, and exfoliazone (Figure 4). 

Whereas it remains to be determined how these compounds initiate activation of the intrinsic pathway and induce alteration of survivin and XIAP leading to sensitization of tumor cells to TRAIL-induced cell death, our results support the idea that marine-derived secondary metabolites are likely to provide novel pharmacologically active drugs, whose discovery and development may lead to therapeutic paradigm changes in oncology, or beyond, for human diseases ranging from auto-immunity to neurodegeneration. 

## Figures and Tables

**Figure 1 cells-09-01760-f001:**
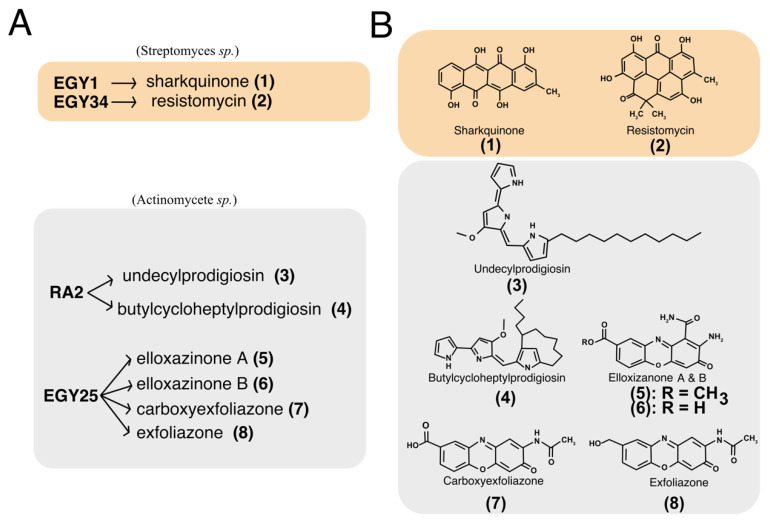
Purified marine actinomycetes-derived secondary metabolites. (**A**) Source of compounds. Sharkquinone (**1**), and resistomycin (**2**) were purified from *Streptomyces* sp. EGY1 and EGY34 isolated from marine sediment of the Red Sea, Egypt. Undecylprodigiosin (**3**) and butylcycloheptylprodigiosin (**4**) were purified from the actinomycetes strain RA2 which was isolated from a marine sponge *Spheciospongia mastoidea* collected in Ras Mohammed, South of Sinai, Egypt. The marine compounds elloxazinone A (**5**), elloxazinone B (**6**), carboxyexfoliazone (**7**), and exfoliazone (**8**) were purified from the actinomycetes strain EGY25, isolated from a sea sand sample obtained from Marsa Matruh city, Mediterranean Sea, Egypt. (**B**) Chemical structures of the corresponding purified marine-derived compounds.

**Figure 2 cells-09-01760-f002:**
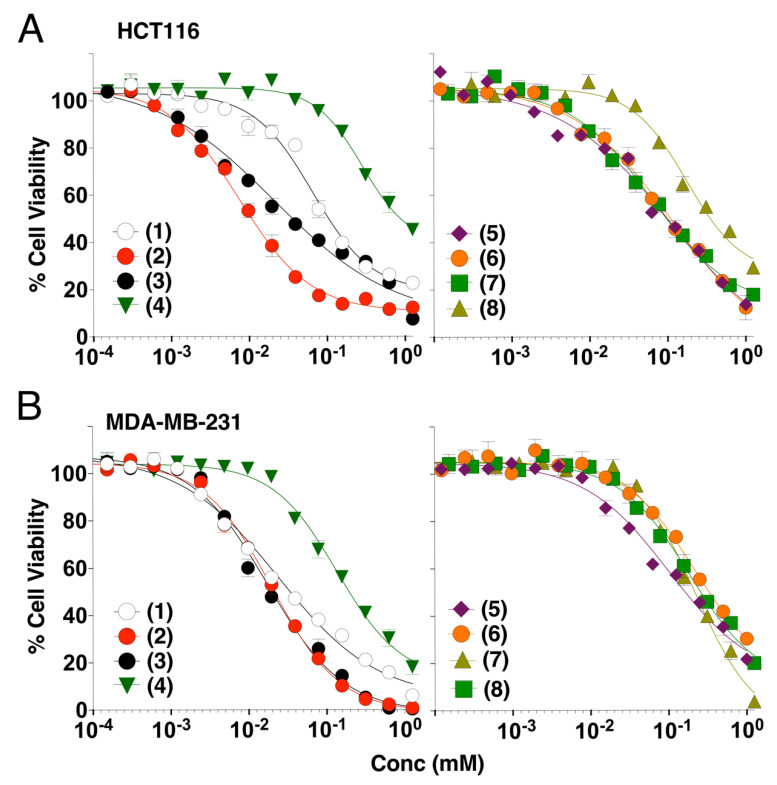
Effect of marine compounds on the viability of HCT116 and MDA-MB-231 cells. (**A**,**B**) The human colorectal and breast cancer cell lines were treated with various concentrations of the marine compounds for 24 h and viability was determined by methylene blue assay. Results correspond to three independent experiments. Error bars represent the SD values. (**1**) Sharkquinone, (**2**) resistomycin, (**3**) undecylprodigiosin, (**4**) butylcycloheptylprodigiosin, (**5**) elloxazinone A, (**6**) elloxazinone B, (**7**) carboxyexfoliazone, and (**8**) exfoliazone.

**Figure 3 cells-09-01760-f003:**
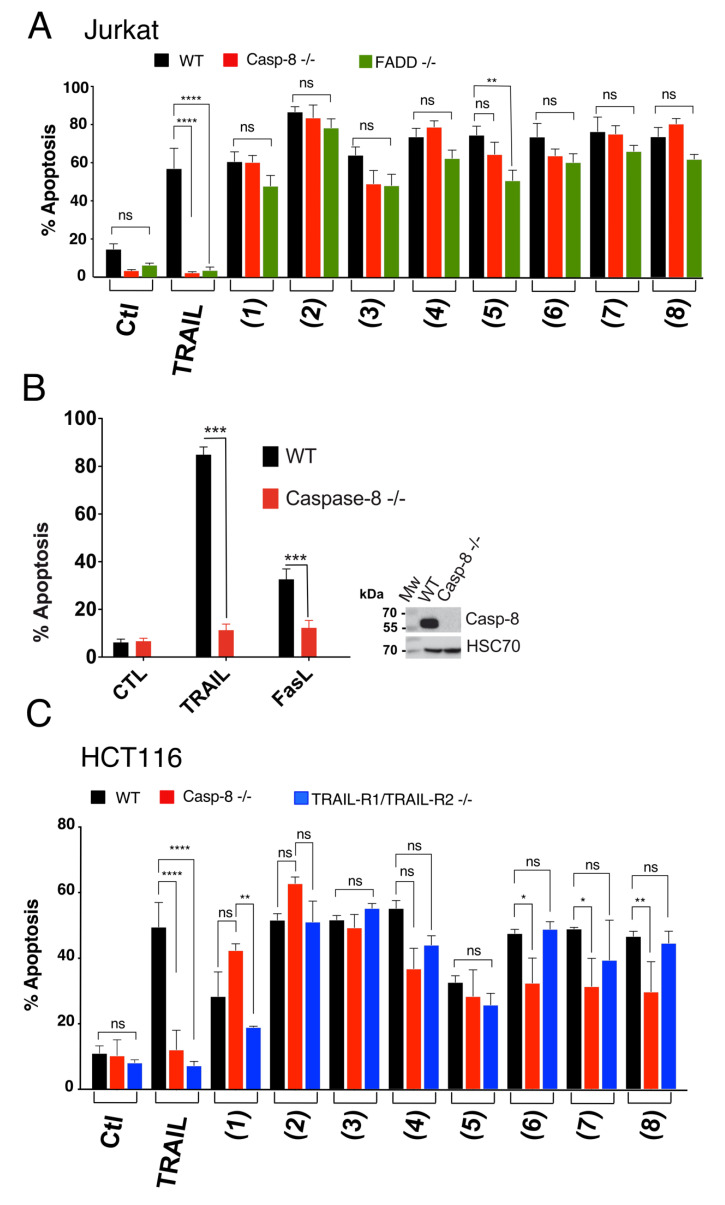
Monitoring apoptosis induced by marine compounds using flow cytometry in Jurkat and HCT116 cells. (**A**) Jurkat WT, Cas-8-/- and FADD-/- cells were treated with the indicated compounds for 24 h then stained with 7AAD and Annexin V and fluorescence was analyzed by flow cytometry. (**B**) Flow cytometry and Western blot analysis to determine the sensitivity of HCT116-Cas8-/- toward TRAIL-and FasL-induced cell death. Cells were treated with and without TRAIL and FasL for 6 h then subjected to dual staining as above and apoptotic percentage was determined by flow cytometry. For Western blot, cell lysates were prepared and loaded on SDS-PAGE (12%) followed by immunoblotting using the corresponding antibodies. HSC70 was used as loading control. (**C**) Colon HCT116 (WT, Cas8-/-, and TRAIL-R1/TRAIL-R2-/-) cancer cell lines were treated as previously mentioned. All values are presented here as ±SD (n = 3). Significance was evaluated by two-way ANOVA. **p* < 0.05, ** *p* < 0.01, *** *p* < 0.001 and **** *p* < 0.0001.

**Figure 4 cells-09-01760-f004:**
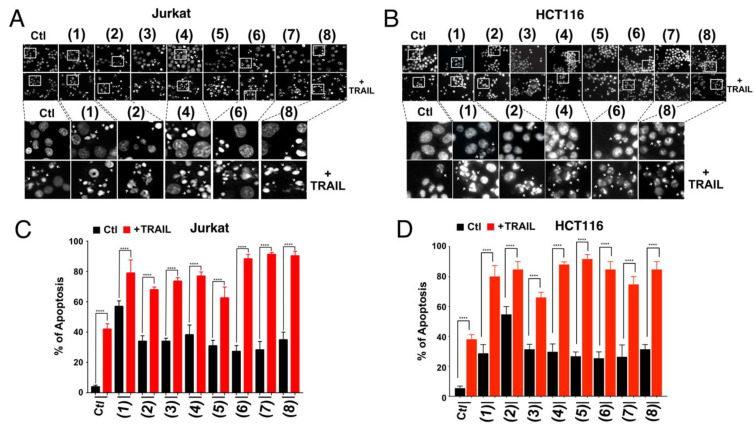
Characterization of apoptosis induced by marine-derived compounds with and without TRAIL in Jurkat and HCT116 cells. (**A**,**B**) Cells were treated with 50 µM of the indicated compounds for 6 h in the absence and in the presence of TRAIL (250 ng) and stained with the fluorescent DNA-binding dye Hoechst 33342. Morphological changes (**A**,**B**) were detected under fluorescent microscope using a blue filter. Corresponding quantifications are shown in (**C**,**D**). All values are presented here as ±SD (n = 3). Significance was evaluated by two-way ANOVA. **** *p* < 0.0001.

**Figure 5 cells-09-01760-f005:**
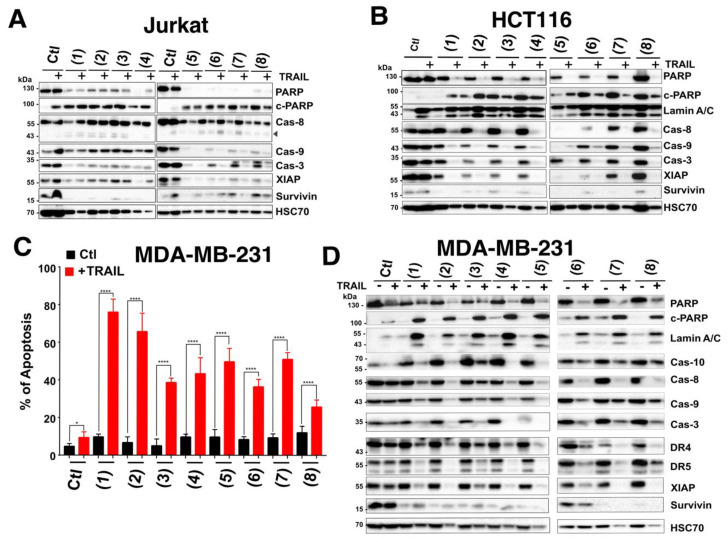
Western blot analysis of pro-apoptotic and anti-apoptotic proteins in cancer cells treated with marine compounds alone or combined with TRAIL. (**A**) Jurkat and (**B**) HCT116 cells were treated or not with marine-derived compounds (50 µM) in the presence or absence of TRAIL (250 ng/mL). Following incubation, cell lysates were prepared and loaded on SDS-PAGE (12%) and the indicated proteins were detected by immunoblotting. (**C**) MDA-MB-231 cells were stimulated as above and apoptosis-induced by marine-derived compounds in the presence or absence of TRAIL was analyzed by Hoechst staining. All values are presented here as ±SD (n = 3). Significance was evaluated by two-way ANOVA. * *p* < 0.05 and **** *p* < 0.0001. (**D**) Corresponding MDA-MB-231 cell lysates were prepared, loaded on SDS-PAGE (12%) and relevant proteins were detected by immunoblot. In all cases, HSC70 was used as a loading control.

**Figure 6 cells-09-01760-f006:**
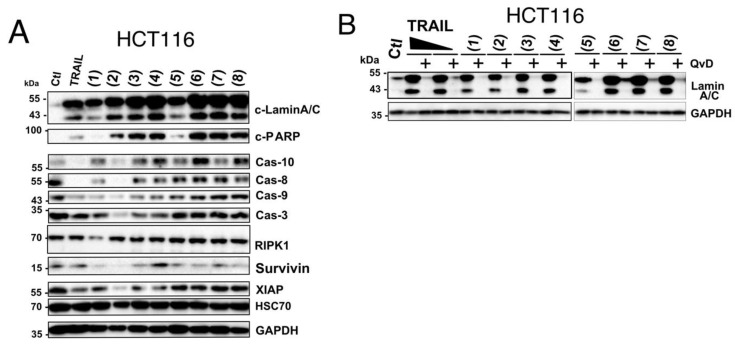
Western blot analysis of pro- and anti-apoptotic proteins in HCT116 cells stimulated with TRAIL or marine compounds. (**A**) Cells were treated with the identified compounds (50 µM) or TRAIL (0.25 µg/mL) for 24 h. Following incubation, cell lysates were prepared and loaded on SDS-PAGE (12%) and the indicated proteins were detected by immunoblotting. GAPDH or HSC70 were used as loading controls. (**B**) Cells were pre-incubated for 30 min in the presence or absence of the *pan*-caspase inhibitor QvD (10 µM) then treated as above with marine compounds (50 µM) or TRAIL (0.25 and 1 µg/mL) for 24 h. Cell lysates were prepared as described above and analyzed for Lamin A/C cleavage by immunoblot.

**Table 1 cells-09-01760-t001:** Half-maximal inhibition concentration (IC_50_) values for marine compounds determined in vitro in the colorectal carcinoma HCT116 and triple negative breast cancer MDA-MB-231 cell lines, determined by methylene blue staining.

Compound	N°	IC_50_ (µM)	
		HCT116 WT	MDA-MB-231
Sharkquinone	1	65.5	24
Resistomycin	2	7.6	19.8
Undecylprodigiosin	3	21.3	17.5
Butylcycloheptylprodigiosin	4	285.8	136.2
Elloxazinone A	5	111.3	25.6
Elloxazinone B	6	114	53.1
Exfoliazone	7	62.3	184.2
Carboxyexfoliazone	8	176.5	221.8

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
