# Peer review of "Marine Actinomycetes-Derived Secondary Metabolites Overcome TRAIL-Resistance via the Intrinsic Pathway through Downregulation of Survivin and XIAP"

_cells, 2020, doi:10.3390/cells9081760_

Round 1

Reviewer 1 Report

In the submitted manuscript, Elmallah and colleagues report on the purification and testing of marine compounds on human tumour cells. They show that these compounds are able to reduce the viability of tumour cells via induction of apoptotic cell death. In addition, these compounds potently sensitized the tumour cells to TRAIL. Although the results are potentially interesting, the quality of all figures showing Western blots results must be improved in the revised version of the manuscript.

Because of several inconsistencies, original blots for all Western blots figures should be presented, either in the supplementary data or at least in the answer to the reviewer comments.

Specific comments:

Figure 3

There are several inconsistencies, which have to be addressed and corrected.

- The marker (70 kDa) in both (A) and (B) suggest that uncleaved (?) forms of Lamin A/C are shown. If this is a mistake, please show both, uncleaved and cleaved forms. In addition, please correct the description of the antibody in the Material and Methods section. According to Santa Cruz, this antibody detects Lamin B1, not Lamin A/C

- Since the used anti Caspase 8 antibody detects all forms, it is not clear why only full-length form has been shown. At least the intermediate cleavage forms, if not all, should be shown (this has been shown for example in the Fig. 6A but again not in 6B or D).

This also holds true for the anti-Caspase 9 antibody – since this antibody detects all forms I wonder why only the disappearance of the full length form is shown, please show also the cleaved forms.

- The used anti-Caspase-3 antibody (Cell signaling, 9661), detects only the cleaved fragments of Caspase-3 and not the full-length protein, however the authors show band corresponding to the full-length form.

- Blots for HSC70 and GAPDH in Fig. 3A are clearly overexposed. Please show shorter exposition. This would allow comparing the gel loading, which seems to be quite unequal in Fig. 3A. The controls shown here do not allow any comparison between the apoptosis-inducing capacities of the tested compounds.

In the main text, the authors have written: “In line with their potential ability to trigger apoptosis, the pan-caspase inhibitor (QvD) prevented both the appearance of Lamin A/C cleaved products (Figure 3B) and apoptosis (Figure 3C and Figure S1), induced both by these compounds and TRAIL.” – Interestingly, there is no Figure 3C at all!

Figure 4

The expression of Caspase 8 (A, B, C), FADD (A), TRAIL-R1/TRAIL-R2 (C) as well as gel loading controls like GAPDH, Actin or HSC70 have to be shown. In the Fig. legend B) it has been even written that HSC70 has been studied, but it is not shown!

Fig. 6

-  Correct gel loading control is missing! Either the compounds themselves induced degradation of HSC70 or different amounts of protein were loaded on the gels for treated and untreated cells. The loading control which does not change between treatments must be provided.

- It is not clear why it was possible to show the  cleavage of Caspase 8 in Jurkat cells but not in HCT116 and MDA-MB-231 cells?

- Is it possible that combining compounds 6, 7, and 8 with TRAIL impaired the cleavage of caspase 8 in MDA-MB-231 cells? or the blots were incorrectly cut?

- In several lines, air bubbles hinder the interpretation of the data.

Minor point

The title of each figure should state shortly what it really shows or what has been studied. It is very confusing when, for example, in s. Fig. 4 only one cell line “Jurkat”  is mentioned in the title and the other one “HCT116” appears later. The same holds true for Fig. 6.

Author Response

Dear Referee,

We would like to apologize for these mistakes, we tried to be as precise as possible, but obviously, some errors remained in the submitted version. We had included a supplementary file in the portal, that is now also available for you at the end of this word file, which includes all uncropped western blotting experiments displayed in our manuscript. Thank you for your vigilance, we have now addressed all your comments, and hope that our answers will satisfy you. Please read the point/point answers provided below.

In the submitted manuscript, Elmallah and colleagues report on the purification and testing of marine compounds on human tumour cells. They show that these compounds are able to reduce the viability of tumour cells via induction of apoptotic cell death. In addition, these compounds potently sensitized the tumour cells to TRAIL. Although the results are potentially interesting, the quality of all figures showing Western blots results must be improved in the revised version of the manuscript.

Because of several inconsistencies, original blots for all Western blots figures should be presented, either in the supplementary data or at least in the answer to the reviewer

Specific comments:

Figure 3 : There are several inconsistencies, which have to be addressed and corrected.

- The marker (70 kDa) in both (A) and (B) suggest that uncleaved (?) forms of Lamin A/C are shown. If this is a mistake, please show both, uncleaved and cleaved forms. In addition, please correct the description of the antibody in the Material and Methods section. According to Santa Cruz, this antibody detects Lamin B1, not Lamin A/C

#1- you are right the molecular weight of cleaved lamin A/C is 50 – 40kDa. It was just a mistyping, we are sorry for that, although the correct size was included in the raw data file of western blot (see also figure for the referee).

#2- Indeed the antibody used here was obtained from Abcam (ab133269). This information is now included in the manuscript.

- Since the used anti Caspase 8 antibody detects all forms, it is not clear why only full-length form has been shown. At least the intermediate cleavage forms, if not all, should be shown (this has been shown for example in the Fig. 6A but again not in 6B or D).

This also holds true for the anti-Caspase 9 antibody – since this antibody detects all forms I wonder why only the disappearance of the full length form is shown, please show also the cleaved forms.

#3- The referee is right, that antibody recognizes both the pro-form and the cleaved product of the caspase-8. While this holds true and is clearly observed in the Jurkat cells, the cleaved products, like caspase-9 cleaved products were not that obvious in the colorectal cancer cell line HCT116 and the MDA-MB-231 cells. The reason why the cleaved caspase 8 fragments (p43) appear in Jurkat (Fig 6A), but not in the other cell lines could be due to cell type. In fact, the detection of these cleaved fragments relies on the quantity of loaded proteins, detergent used to extract proteins, kinetic of the stimulation and on the cell line, but also on the batch of antibody that is obtained from the provider, and as you probably know they are not always reliable. That is the reason why we have also used antibodies to detect caspase substrate cleavage products, such as Lamin A/C or PARP whose cleaved products, on the contrary accumulate in the lysates, balancing loading issues. Their appearance together with the demonstration that the pan-caspase inhibitor (qVD) impairs the pro-apoptotic signaling pathway activated by our compounds, is in our opinion, sufficient to conclude that the cell death program through activation of caspases.

- The used anti-Caspase-3 antibody (Cell signaling, 9661), detects only the cleaved fragments of Caspase-3 and not the full-length protein, however the authors show band corresponding to the full-length form.

#4- We are deeply sorry for this mistake, in fact we have used the anti-Caspase-3 antibody (Cell signaling, 9665), which recognizes the proform, but was also described as able to recognize the p17. Yet, this was seldom the case in our hands, and the antibody is not sold anymore by Cell signaling, probably due to this reason. We have corrected the reference number in the manuscript. Again, we didn’t rely solely on caspase-3 cleaved products but also on the appearance of PARP cleaved products (c-PARP), a direct substrate of the caspase-3, as well as on nuclear fragmentation as evidenced by Hoechst staining, which give a clear evidence for the activation of caspase-3. 

- Blots for HSC70 and GAPDH in Fig. 3A are clearly overexposed. Please show shorter exposition. This would allow comparing the gel loading, which seems to be quite unequal in Fig. 3A. The controls shown here do not allow any comparison between the apoptosis-inducing capacities of the tested compounds.

#5- The referee is right, the loading was sometimes uneasy to handle due to a) massive cell death that was induced in these cells and b) to the need to work with limited number of cells to spare our precious compounds. In the case of these two house-keeping genes, the exposition time shown is the lowest. While we agree that the loading controls are not perfect, we shall stress here that due to limitations of the compounds, we had to scale down our experiments to spare them. The immunoblots are, nonetheless of sufficient quality to draw the conclusions that we have proposed in the manuscript. We actually paid a lot of attention not to over-sale our findings.

In the main text, the authors have written: “In line with their potential ability to trigger apoptosis, the pan-caspase inhibitor (QvD) prevented both the appearance of Lamin A/C cleaved products (Figure 3B) and apoptosis (Figure 3C and Figure S1), induced both by these compounds and TRAIL.” – Interestingly, there is no Figure 3C at all!

#6- Sorry for this mistake. Originally the flow cytometric histograms were presented in this figure, but were next transferred as a supplementary figure (Fig S1). We have now corrected this mistake.

Figure 4

The expression of Caspase 8 (A, B, C), FADD (A), TRAIL-R1/TRAIL-R2 (C) as well as gel loading controls like GAPDH, Actin or HSC70 have to be shown. In the Fig. legend B) it has been even written that HSC70 has been studied, but it is not shown!

#7- Dear referee, you are right these blots are missing. Unfortunately, the post-doc who did these experiments didn’t provide these blots and has now left. With the COVID, I must say that it will be difficult to get these results fast. Yet we have been working for a long time with these cells which are ver­­­y well characterized (See Figure for Referee 1A-C, page 4). We have relied on the loss of TRAIL sensitivity for the FADD-/- and Caspase-8-/- Jurkat cells. We have included the profile of these cells obtained when I was a post-doc in Jurg Tschopp’s laboratory (Bodmer Nat Cell 2000; and Holler Nat Immunol 2001). These where originally obtained by Juo and Blenis as commented in the manuscript.

For HCT116 caspase-8-/- cells the blot is shown and we actually considered that FADD would serve as a house-keeping protein for loading control. This has now been corrected with HSC70 both in the figure and in the legend text. Again, for the TRAIL receptor deficient cells, we work on a regular basis with the latter and these have been extensively described in our manuscript (Dufour et al; Oncotarget 2017, see also figure 1 for the referees). We thus believe that showing the loss of expression of DR4 and DR5 is not needed. But should you like us to include this we would happily do it.

Fig. 6

-  Correct gel loading control is missing! Either the compounds themselves induced degradation of HSC70 or different amounts of protein were loaded on the gels for treated and untreated cells. The loading control which does not change between treatments must be provided.

- It is not clear why it was possible to show the  cleavage of Caspase 8 in Jurkat cells but not in HCT116 and MDA-MB-231 cells?

 - Is it possible that combining compounds 6, 7, and 8 with TRAIL impaired the cleavage of caspase 8 in MDA-MB-231 cells? or the blots were incorrectly cut?

- In several lines, air bubbles hinder the interpretation of the data.

#8- We agree with you that there was a mistake in the figure version that was supplied with the document and are sorry for this. To answer your point, we had prepared a figure showing all raw data (see page 5). Part of the answers are given, above #Answers 3/#4 and #5. Likewise, although the loading of samples obtained from several compounds was clearly lower than controls or other compounds, appearance of Lamin A/C or PARP cleaved products showed that apoptosis was ongoing. We had to spare our compounds and unfortunately will not be able to redo these experiments. Yet the overall conclusion is not overstated and we paid attention to this.

With respect to Jurkat cells, the appearance of cleaved caspase-8 fragments, although detectable, is much lower than expected. In our opinion it is likely related to the batch N° of the caspase-8 antibody, given that the cleaved products of downstream caspase substrates are easily detected and that the proform of C8, C9 and C3 clearly disappear after stimulation in a number of conditions.  

We again apologize for the mistake figure 6D, for the caspase-8, this has now been corrected, see also raw data.

Minor point

The title of each figure should state shortly what it really shows or what has been studied. It is very confusing when, for example, in s. Fig. 4 only one cell line “Jurkat”  is mentioned in the title and the other one “HCT116” appears later. The same holds true for Fig. 6.

We understand that this may be an issue and have thus made some corrections of the legends accordingly.

Thank you for your insightful comments.

Figure for the referee (see Page 4 and 5, attached document)

Extra figure to be uploaded to Cells’s portal

Reviewer 2 Report

Elamallah et al. reported that purified secondary metabolites derived from marine Actinomycetes exert anti-tumor activity and sensitizes cancer cells to TRAIL-induced apoptosis, probably by suppressing survivin and XIAP expression in HCT116 and MDA-MB-231 cancer cells. The anti-cancer effects of these secondary metabolites are apparent, and the authors addressed the underlying mechanisms well. However, there are still some concerns about the results to draw the conclusions.

Major concern:

1          To analyze molecular mechanisms and the combination effects of these compounds with TRAIL, the authors used the same concentration (50 µM) for all the compounds. Please explain the reason why the concentration (50 µM) was chosen for all of them.

2          All these compounds suppressed the expression of survivin and XIAP in Jurkat, HCT116, and MDA-MB-231 cancer cells. Although the results are impressive, it is still uncertain whether these IAPs are involved in cell death and sensitization effects by these compounds. Please confirm this point, e.g., by using siRNA or CRISP/Cas9 against survivin and XIAP, or pharmacological inhibition of survivin (YM155, etc.). Alternatively, if you do not perform further experiments, please carefully discuss the point.

Minor points

1          In Figure 2A and 2B, although the X-axis (concentration) looks logarithmic scale, it is not a logarithmic scale. The number of short tick marks on the X-axis (concentration) is wrong. The number of short tick marks between two long tick marks should be eight, not nine. Furthermore, the distance between short tick marks should not be even.

2          Please replace 'surviving' in the abstract with 'survivin'.

Author Response

Dear Referee,

We would like to thank you for your comments. Please find below our point/point answers. We have also included a figure for your perusal, and made corrections in the revised version of our manuscript, accordingly.

Comments and Suggestions for Authors

Elmallah et al. reported that purified secondary metabolites derived from marine Actinomycetes exert anti-tumor activity and sensitizes cancer cells to TRAIL-induced apoptosis, probably by suppressing survivin and XIAP expression in HCT116 and MDA-MB-231 cancer cells. The anti-cancer effects of these secondary metabolites are apparent, and the authors addressed the underlying mechanisms well. However, there are still some concerns about the results to draw the conclusions.

Major concern:

1          To analyze molecular mechanisms and the combination effects of these compounds with TRAIL, the authors used the same concentration (50 µM) for all the compounds. Please explain the reason why the concentration (50 µM) was chosen for all of them.

We have fixed the concentration of compounds (50 µM) according to the compound screening that we had performed using flow cytometry (annexin V staining) and Hoescht staining. The referee is right, compared to table 1 the fixed chosen concentration of 50 µM appears odd, but he or she has to consider that these IC50 calculations were based on the methylene blue (MB) assay. This assay allows the quantification of cells remaining attached to the 96-well plate, based on the property of the blue dye to interact with proteins. Thus, the difference of staining does not only reflect the loss of cell viability, but can also result from changes in the metabolic status of the cells or adherence, regardless of cytotoxicity or apoptosis. In agreement with a possible bias of the assay, and as compared to the calculated IC50, based on apoptosis assays as monitored by annexinV or Hoechst staining (in a double-blind fashion), as shown in figure (4, 5 and 6), the chosen concentration corresponds almost to a concentration that induces apoptosis in around 50% of the cells, except for MDA-MB-231, that display a greater resistance (Figure 6). We had a really hard time to work with several of these compounds due to autofluorescence, by flow cytometry (Facs, see figure for the referee page 3). These issues were commented in the manuscript and shown as supplementary figures. That is why we also performed Hoechst staining to quantify apoptosis based on morphological changes. You will likely appreciate, that the rate of apoptosis, as assessed by Hoechst staining and shown figure 5 and 6, are even slightly lower than those obtained by annexin V (Facs). Therefore, and in order to avoid any confusion, we have mentioned in the text that the IC50 calculated and shown table 2, reflects the results obtained by MB and corresponds to this particular experimental setting.

2          All these compounds suppressed the expression of survivin and XIAP in Jurkat, HCT116, and MDA-MB-231 cancer cells. Although the results are impressive, it is still uncertain whether these IAPs are involved in cell death and sensitization effects by these compounds. Please confirm this point, e.g., by using siRNA or CRISP/Cas9 against survivin and XIAP, or pharmacological inhibition of survivin (YM155, etc.). Alternatively, if you do not perform further experiments, please carefully discuss the point.

We thank the referee for this interesting question. Indeed, loss of XIAP or survivin, including its inhibition by YM155 is reported in a large number of research articles to increase TRAIL-induced cell death. Two of them even provide the demonstration that YM155 sensitizes triple negative breast cancers to TRAIL-induced cell death (Pennati et al., 2015), including MDA-MB-231 cells (Woo et al., 2016).  We have now discussed this point in the revised version of the manuscript (See discussion).

Minor points

1          In Figure 2A and 2B, although the X-axis (concentration) looks logarithmic scale, it is not a logarithmic scale. The number of short tick marks on the X-axis (concentration) is wrong. The number of short tick marks between two long tick marks should be eight, not nine. Furthermore, the distance between short tick marks should not be even.

Dear referee, we have highlighted the major ticks in the figure. Plots were created using GraphPad Prism 8, and the log10 logarithmic scale, which displays 9 minors ticks.

2          Please replace 'surviving' in the abstract with 'survivin'.

Thanks a lot, we didn’t even notice this typo, that often occurs due to the automatic spelling/grammar corrections of the software.

References:

Pennati, M., Sbarra, S., De Cesare, M., Lopergolo, A., Locatelli, S.L., Campi, E., Daidone, M.G., Carlo-Stella, C., Gianni, A.M., and Zaffaroni, N. (2015). YM155 sensitizes triple-negative breast cancer to membrane-bound TRAIL through p38 MAPK- and CHOP-mediated DR5 upregulation. Int J Cancer 136, 299-309.

Woo, S.M., Min, K.J., Seo, B.R., and Kwon, T.K. (2016). YM155 sensitizes TRAIL-induced apoptosis through cathepsin S-dependent down-regulation of Mcl-1 and NF-kappaB-mediated down-regulation of c-FLIP expression in human renal carcinoma Caki cells. Oncotarget 7, 61520-61532.

Many thanks for your insightful comments.

Round 2

Reviewer 1 Report

Thank you for your detailed and straitforward answer. Although some of my initial requests are still not fulfilled, your response and arguments are convincing.

I suggest to add all the presented uncropped Western blots in the final Supplementary data files.